# Mechanical constraints to unbound expansion of *B. subtilis* on semi-solid surfaces

Mojca Krajnc,[1] Chenyi Fei,[2] Andrej Košmrlj,[3,4] Mitjan Kalin,[5] David Stopar[1]

**ABSTRACT** The effects of surface mechanical constraints that may promote or prevent bacterial expansion on semi-solid surfaces are largely unknown. In this work, we have manufactured agar surfaces with different viscoelasticity, topography, and roughness. To capture the essential biophysics of the bacterial expansion we have developed a continuum model that faithfully reproduces the main patterns of the short-range and long-range expansion with two critical parameters: local interfacial forces and colony viscosity. Cohesive energy of the bacterial colony that determines the extent of exploration was dependent on agar surface viscoelasticity. On soft surfaces, bacteria produce low viscoelastic colonies that allow guided population of bacteria to traverse distances that are six orders of magnitude larger than the size of the individual bacterium. Bacteria growing on stiff surfaces produce colonies with significantly increased viscoelasticity that prevent bacterial exploration of new territory and allow formation of a very steep cliff at the edge of the colony. Upon flooding of the rough surfaces, we have induced aquaplaning and spreading of bacteria. A layer of water between the bacterium and surface results in a loss of traction allowing bacteria to spread across the otherwise inhibitory rough surface. The results shed new light on the bacterial ability to rapidly colonize new territories.

**IMPORTANCE** How bacterial cells colonize new territory is a problem of fundamental microbiological and biophysical interest and is key to the emergence of several phenomena of biological, ecological, and medical relevance. Here, we demonstrate how bacteria stuck in a colony of finite size can resume exploration of new territory by aquaplaning and how they fine tune biofilm viscoelasticity to surface material properties that allows them differential mobility. We show how changing local interfacial forces and colony viscosity results in a plethora of bacterial morphologies on surfaces with different physical and mechanical properties.

**KEYWORDS** *B. subtilis*, expansion, surface topography, roughness, viscoelasticity, mathematical modeling

W hy bacterial colonies on semi-solid surfaces usually have a finite size in spite of unbounded growth potential is not well understood. This is a problem of fundamental microbiological and biophysical interest and is key to the emergence of several phenomena of biological, ecological, and medical relevance. The effects of surface mechanical constraints (i.e., viscoelasticity, roughness, and friction) that may promote or prevent bacterial expansion are largely unknown. The exploration of new territory is used by bacteria to locate new resources (1, 2), acquire new surfaces (3), escape unfavorable conditions (4, 5), and avoid competition (6–11).

Previously, some of the authors have considered mechanical deformations of substrate and friction between bacteria and substrate to model the morphogenesis of growing bacterial colonies (12). Distinct spatiotemporal patterns were observed in the

Address correspondence to David Stopar, david.stopar@bf.uni-lj.si.

The authors declare no conflict of interest.

See the funding table on p. 14.

bacterial colonies grown on agar surfaces that were dependent on the friction coefficient. Several key factors have been recognized that determine friction force between bacteria and surface, such as cell-substrate interactions (13–15), surface wetness (1, 16–18), surface roughness (14, 19–21), and surface energy (22–25). To promote exploration success bacteria may alter their physical environment by releasing surfactants or extracellular polymers (7, 26–29). It is, however, intriguing that production of surfactants on rough surfaces does not allow exploration of new territory suggesting that other physical and mechanical surface properties are important for bacterial mobility.

Long-range expansion by dendrite outgrowth can occur in *Bacillus subtilis* and several other bacterial species only on soft surfaces (16, 30–35). In *B. subtilis*, dendrites expand from the mother colony at a constant rate (up to 10 mm h$^{-1}$) and go through a series of distinct morphologically and genetically defined stages (36–38). At the tip of the growing dendrite (1–2 mm) a dense and homogenous population of hyperflagellated explorer cells resides which exhibit hypermotility, have a reduced cell size, high rate of DNA, protein and cell wall synthesis, and a very high multiplication rate (39). It has been proposed that as the explorer cells divide one of the daughter cells remains an explorer cell whereas the other differentiates into a non-motile, immobilized, low metabolic settler cell. There is a progressive shutdown of bacterial activity from the tip to the base of the dendrite (39). As long-range dendritic growth is observed only on soft surfaces, the major unresolved issue is this: what are the mechanical constraints that prevent exploration on stiffer and rougher surfaces?

In this work, we show how surface mechanical properties guide the expansion of *B. subtilis* and how it can be modulated. We have manufactured agar surfaces with different viscoelasticity, topology, and roughness. We have induced expansion by aquaplaning in mutant strains with defects in the expansion on surfaces that were previously believed to be inhibitory for the exploration, or inhibited the expansion by increased roughness and genetic modification of extracellular matrix components production. We have measured how bacterial biofilm viscoelastic properties change when grown on surfaces with different stiffness. To capture the essential physics of the long-range bacterial expansion a new continuum model was developed.

## RESULTS

### Short-range and long-range bacterial expansion

The short-range and long-range expansion of the wild-type *B. subtilis* PS-216 strain in the MSgg medium at different nutrient and agar concentrations is shown in Fig. 1. The extent of the expansion was estimated from the area occupied by the explorer cells (Fig. S1 in Supporting Information Appendix). On plates with agar concentrations below 1%, long-range dendrite expansion was the dominant form of bacterial expansion. The width of dendrites increased with increasing nutrient concentration, occasionally neighboring dendrites merged. The exploration of the new territory decreased dramatically with increasing agar concentration. At agar concentrations of 1.5% or higher, the exploration was limited to a few bud structures (orange arrows) emerging from the edge of the mother colony. With increasing nutrient concentration, a tendency for a short-range expansion with a collar of densely packed bacteria around the mother colony (green arrows) was observed.

The expansion, as defined in this work, is characterized by the growth of bacteria beyond the perimeter of the mother colony (i.e., beyond the area of the inoculum). Inability to explore new territory at high agar concentration induced physical stress on the growing bacterial population in the mother colony which was resolved through mechanical instabilities (wrinkle formation). The density and magnitude of wrinkles in the mother colony increased with nutrient concentration. In contrast, at low-agar concentrations, the physical stress within the mother colony could be resolved at several locations around the perimeter of the mother colony by dendrite outgrowth.

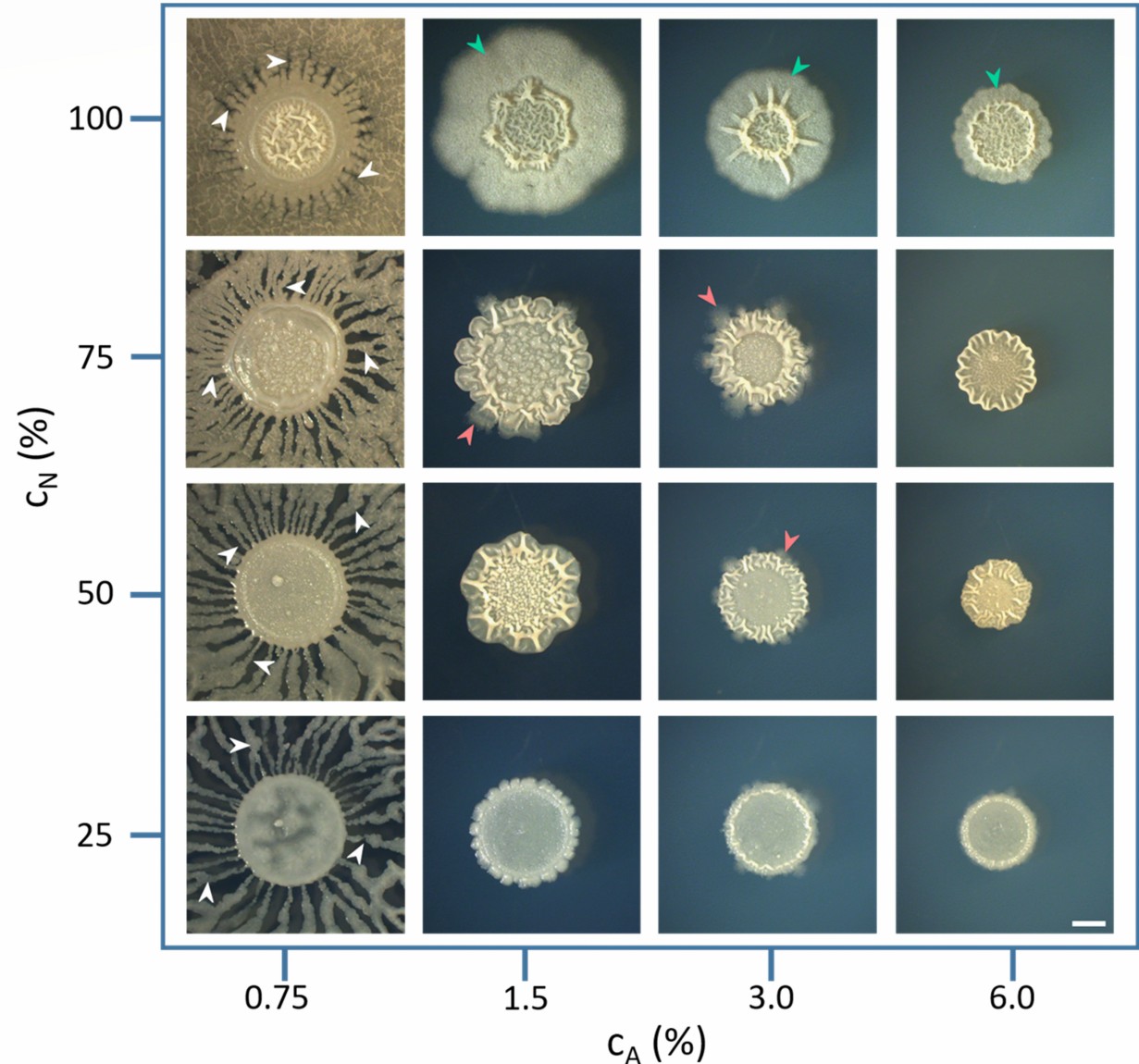

**FIG 1** The bacterial expansion of *B. subtilis* PS-216 wt strain on MSgg growth medium with different nutrient ($C_N$) and agar ($C_A$) concentrations. The scale bar is the same for all images and represents 2 mm. White arrows represent long-range dendritic growth, green arrows represent short-range expansion, and orange arrows represent outbursts from the mother colony. The expansion was mostly observed at low-agar/high-nutrient concentrations.

## Essential physics of bacterial expansion

To capture the essential physics of the observed bacterial expansion patterns on different agar surfaces, we have developed a continuum model that considers the growth of the bacterial colony, the buildup of mechanical stress in the colony, the local surfactin-induced Marangoni force, and the friction between the colony and the agar substrate (12, 40). The simulated morphologies of the bacterial colony as a function of surface friction and colony viscosity are given in Fig. 2. The model faithfully reproduces the branching morphologies of the experimental colonies on different surfaces with two essential physical parameters (viscosity of the bacterial colony and local interfacial force). The simulation suggests a sharp transition from long-range to short-range expansion with an increase in bacterial colony viscosity as experimentally observed. With the increase of viscosity, bacteria could only form bud structures around the mother colony

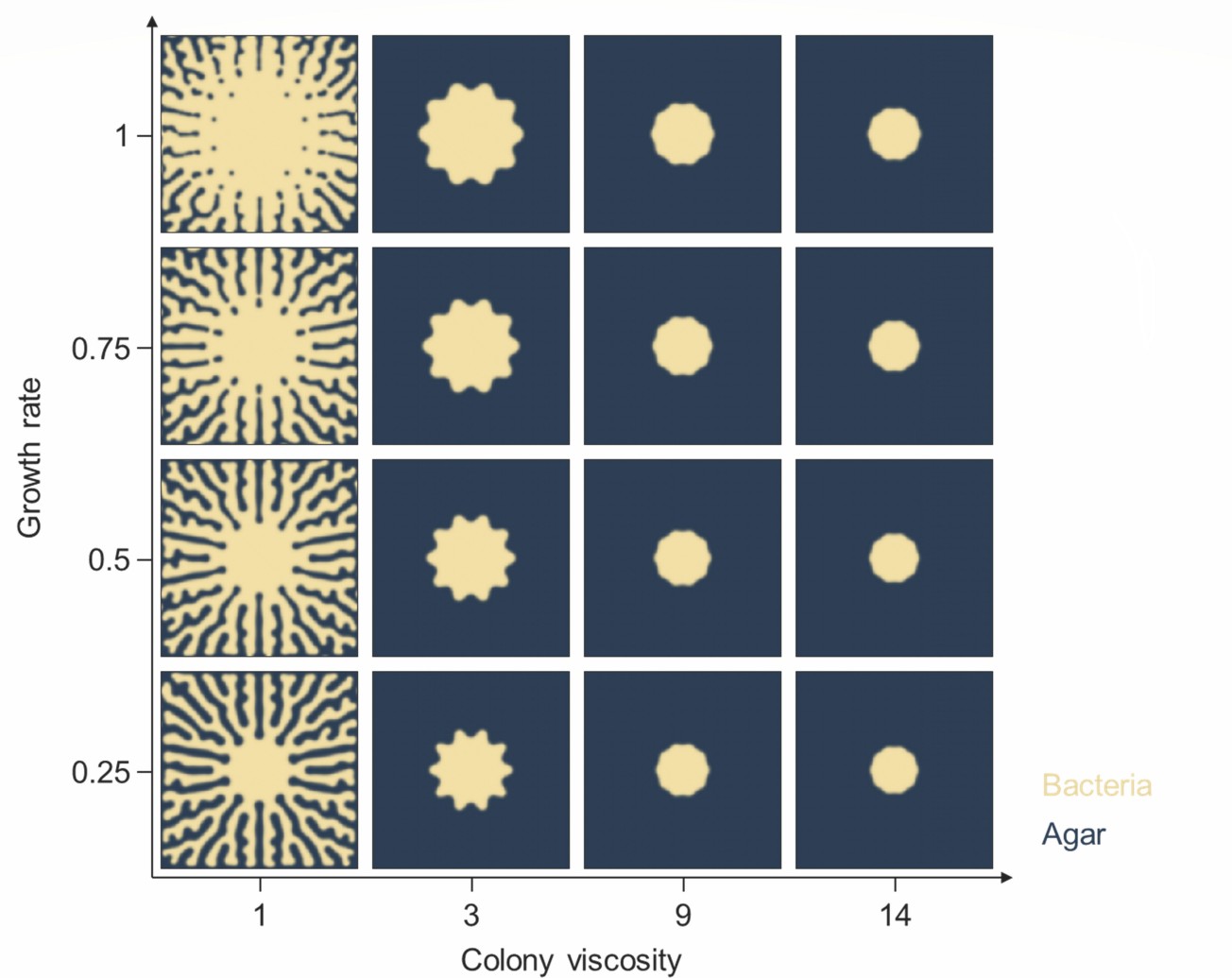

**FIG 2** The simulation of the bacterial short-range and long-range expansion. The continuum model explicitly considers the growth rate of bacteria, the buildup of mechanical stress in the colony, the Marangoni force, the friction between the colony and the agar substrate, and agar surface viscosity.

but not dendrites. Increasing growth rate increases the size of the mother colony, but cannot instigate long-range bacterial expansion.

To correctly capture the branching morphologies on low friction surfaces, one needs to explicitly simulate the surfactin concentration field (see Materials and Methods). Surfactin is a prerequisite for expansion, a fact that has been experimentally verified with surfactin deficient mutant (ΔsrfA), where neither long-range nor short-range expansion was observed on a smooth surface (Fig. S2 in Supporting Information Appendix). It is important to note that surfactin production, which is indispensable for dendrite expansion on soft surfaces and low bacterial colony viscosities, cannot facilitate dendrite branching on stiffer surfaces (Fig. 1) suggesting that other physio-chemical factors determine the exploration of new territory.

We hypothesized that the mechanical properties of the biofilm materials could play predominant roles in driving bacterial exploratory behaviors. To test this hypothesis, we measured the viscoelastic properties of the agar and the biofilms at different agar concentrations (Fig. 3). Agar elastic modulus varied linearly between 2 and 40 kPa with the agar concentration. The yield point of different agars had a sharp transition around the agar concentration of 1.5% (Fig. S3 in Supporting Information Appendix), which correlated with a transition from long-range to short-range expansion.

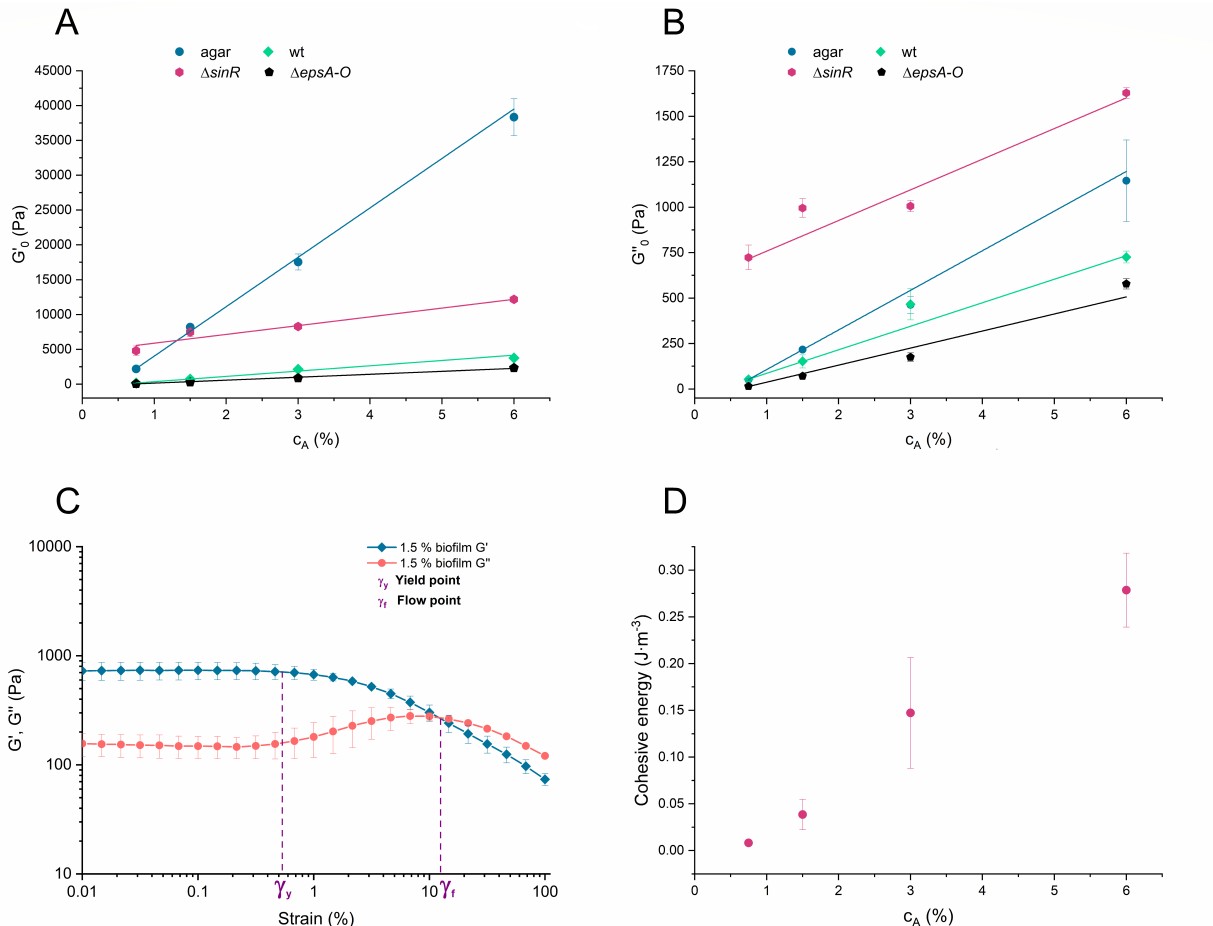

**FIG 3** Viscoelastic properties of agar and bacterial biofilms grown on different agar concentrations for wild-type colonies and several mutants with either overproduction (Δ*sinR*) or impaired (Δ*epsA-O*) production of extracellular matrix components. (A and B) The storage $G'$ (A) and loss moduli $G''$ (B) as a function of agar concentration. (C) Yield point ($\gamma_c$) and flow point ($\gamma_f$) indicated on viscoelastic curves for the biofilm grown on 1.5% agar. (D) Cohesive energy density calculated as $(1/2)\gamma_{crit}^2 G'$. Results are presented as mean ± standard deviation from three independent biological measurements.

The biofilms grown on soft agars have low-yielding points and a large flow point which indicate that biofilm material grown on soft agar is highly deformable and does not crack easily (Fig. 3C). This allows bacteria to stick together and move as an ensemble over the surface. Biofilms grown on stiffer agars had high elasticity and viscosity moduli which could inhibit bacterial spreading. An increase in biofilms cohesion energy density when grown on stiffer agars correlated with a significant increase of expression of *epsA-O* operon (Fig. S4 in Supporting Information Appendix), which suggests that the production of EpsA-O extracellular polysaccharide contributes to colony viscoelasticity. The increased expression of EpsA-O polysaccharide at higher agar concentrations correlates with biofilm formation several layers thick, as opposed to the monolayer of mobile cells on a surface with a low concentration of agar. Even after a prolonged incubation bacterial lateral expansion was halted.

To prove that modulation of extracellular polymer production can indeed modify biofilm viscoelasticity, and hence the expansion, we have used mutants that either overproduce the adhesive extracellular material (Δ*sinR*) or mutants that do not produce extracellular polysaccharides (Δ*epsA-O*). As expected, the viscoelastic moduli $G'$ and G″ of Δ*sinR* biofilms were much larger than those of the wild type (Fig. 3). Consistent with the model prediction, the overproduction of the adhesive extracellular material completely inhibited expansion (Fig. S2 in Supporting Information Appendix). On the other hand, the absence of EpsA-O exopolysaccharides made the bacterial colony softer, less viscous,

increased local spreading of the exploratory cells, and allowed better filling of the space between the dendrites compared to the wild type.

## Surface roughness guides bacterial expansion

As friction force between bacterium and surface, which determines bacterial expansion, is dependent on surface topography and surface roughness, we have measured several agar surface roughness parameters. At 0.75% agar concentration, the agar surface was smooth with nano-roughness in the range of tens of nanometers ($S_a \approx 0.04$ µm). This is approximately two orders of magnitude smaller than the size of a typical bacterial cell and does not provide a hurdle for bacterial spreading. With increasing agar concentration, the average roughness increased for several orders of magnitude up to $S_a \approx 10$ µm on 6% agar, which is comparable to the size of bacteria (peak-to-valley heights are even larger) and can impact bacterial spreading. Although agar is used routinely in microbiological laboratories, it is not common knowledge that agar surface roughness can change dramatically in a usually used range of concentrations (i.e., from 0.75% to 6%). To further characterize the roughness of different agar surfaces, we have determined surface moments skewness and kurtosis of the observed agar surfaces (Fig. S5 in Supporting Information Appendix). With increasing agar concentration, skewness becomes progressively more negative indicating a proportional increase of pit structures, which are, based on kurtosis values, becoming deeper and narrower. This implies that spreading over rougher agar surfaces should increase friction and slow down bacterial advancement. An attempt has been made to measure the friction coefficient directly with different AFM (atomic force microscopy) probes. Due to the soft nature of agar gels, it was not possible to accurately determine the friction coefficient. Even at nN normal load, the AFM probe sinks into the soft agar and plows through the agar.

To observe how bacteria negotiate significant differences in agar surface roughness across the multiple length scales, we have grown bacteria on different agar surfaces and measured biofilm topography from the center of the mother colony to the tip of the dendrite. At the tip of the dendrite at 0.75% agar concentration (Fig. 4A), bacteria formed extended finger structures. The unexplored agar surface ahead and between the fingers had nano-roughness. The bacterial colony rises gradually from the smooth agar surface. Ahead of the first bacterial cells (up to 100 µm, seen as a light blue color), the agar was flooded with extracellular material. It is expected that surfactin has been excreted ahead of bacterial cells (41); however, a rather thick layer of excreted material ahead of the colony would argue that in addition to surfactin, extracellular material has been secreted as well which lubricates the advancement of the bacterial colony. The bacterial colony gradually increased in thickness with a slope of ≈2°. This indicates that the majority of cells in the dendrite fingers are single-layered. The single layer extends up to 500 µm inside the finger. The side of the finger was wavy with many bays and coves. Theoretically, there should be no limit to long-range exploration on the surface with no mechanical constraints (Fig. S6 in Supporting Information Appendix). For example, when bacteria were inoculated at one end of the 1.2-m tube, they rapidly (in <20 h) explore the entire lane. This is an enormous distance on a bacterial spatial scale and is combined with a phenomenal bacterial colony exploration average speed of ≈5 body lengths per second.

At increased agar concentrations, bacteria explore new territory only locally (Fig. 4B). At the forefront, a wall of layered bacteria was followed by a relatively smooth layer of the extracellular material, which was part of a larger extracellular lagoon system with bacterial peninsulas, bays, and coves. The lowland with lagoons and islands of bacterial cells extended for ≈150–250 µm inland when a transition to a thicker multilayer bacterial colony occurred with a rough colony surface. The unoccupied agar ahead of the colony had an underlying wave structure with an amplitude of up to 150 nm. There was no observable extracellular matrix secretion ahead of the bacterial colony.

A dramatically different edge of the bacterial colony was observed on 6% agar (Fig. 4C). The edge of the biofilm was very steep. A sharp cliff (up to 100 µm high) demarcates

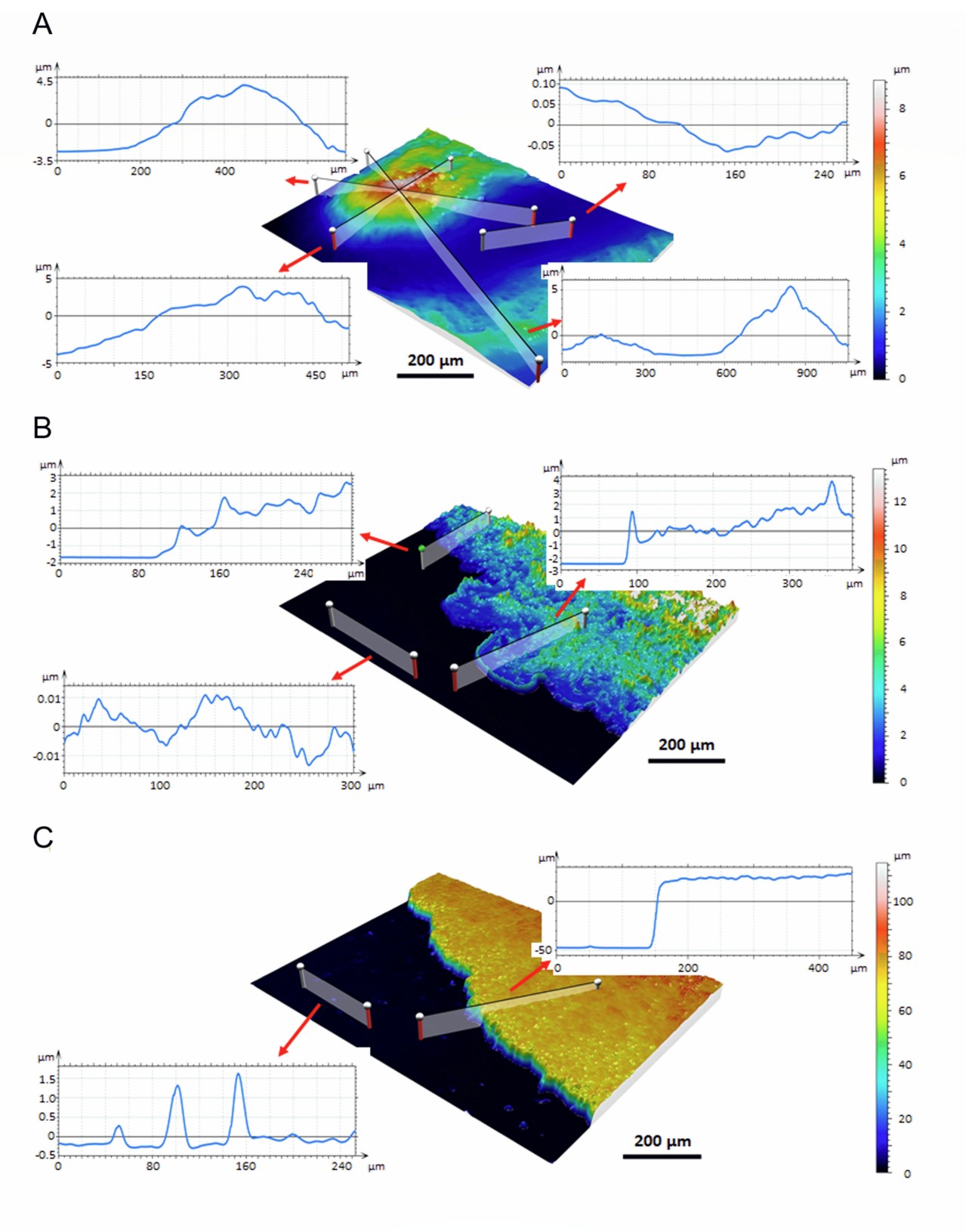

**FIG 4** Biofilm and agar topography on different agar surfaces. Surface 2D profiles are given at the tip of the dendrite branch on 0.75% agar (A), at the edge of the short-range bacterial expansion on 1.5% (B), and at the edge of the mother colony on 6% agar (C). Two-dimensional surface profiles in different directions in the agar and colony are given.

a transition from a plateau of bacteria in the colony to the agar surface. The slope of the cliff varied from 40° to 65°. Such an angle of repose suggests a rather strong cohesive energy between bacteria in the mother colony implying very poor flow characteristics (42, 43). The increased cohesive energy of biofilm grown on 6% agar has been experimentally verified (Fig. 3). The unoccupied agar surface was rough and contained many mounds and holes. Several obstacles with the size on the order of the size of bacteria were visible. There was no layer of extracellular material secreted ahead of the colony although both *srfA* and *epsA-O* expressions increased significantly in the mother colony (Fig. S4 in Supporting Information Appendix).

## Aquaplaning of bacteria on rough surfaces

As increased roughness is a major obstacle to bacterial spreading, we have checked if artificially decreased roughness relaxes mechanical constraints for bacterial spreading. To do so, we flooded the surface around the stalled colony of Δ*srfA* mutant with a drop of water containing surfactin (Fig. 5). Upon addition of surfactin solution, the short-range expansion was observed in all directions to the edge of the new wetted front. On all agar concentrations, bacteria explore new territory with approximately the same speed suggesting that surface properties were the same. To convince ourselves that

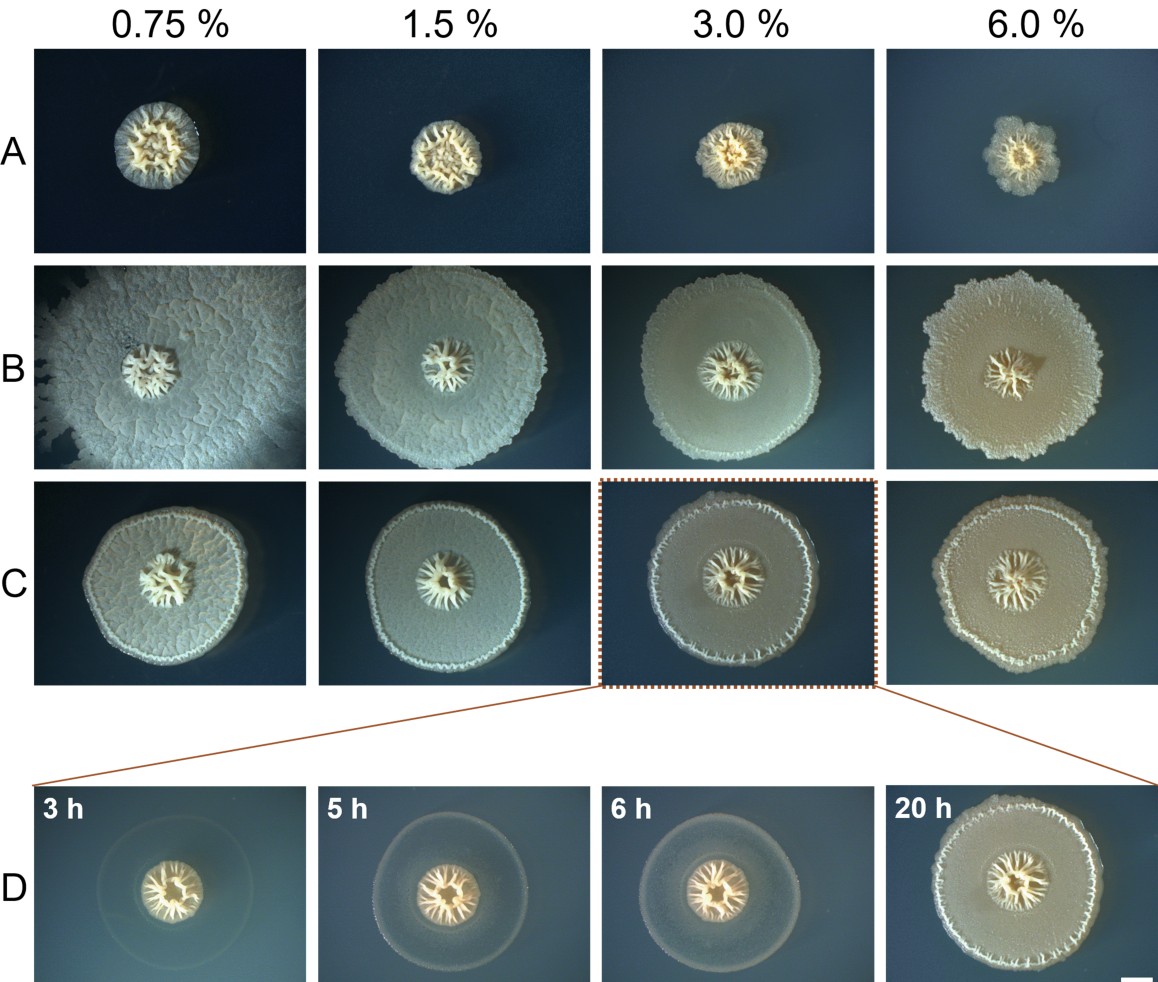

**FIG 5** Induction of the bacterial expansion and aquaplaning. (A) The growth of surfactin mutant (Δ*srfA*) on different agar concentrations after 40 h of incubation. The colonies of Δ*srfA* were grown for 20 h when surfactin sodium salt solution (B) or MiliQ water (C) was added around the mother colony and incubated for additional 20 h. (D) Time lapse of the colony (panel C at 3.0% in Fig. 5) after adding MiliQ water. Scale bar represents 2 mm and is the same for all images.

reduced surface roughness and not surfactin was the causative agent for the decrease of the expansion barrier, we added MiliQ water around the mother colony. The effects of adding MiliQ water were remarkably similar to the effect of the surfactin (Fig. 5) except on 0.75% agar where surfactin allowed morphologically different short-range exploration in addition to long-range exploration of new territory. This suggests that surfactin increases wetting only on smooth surfaces but has no additional effect on rougher surfaces. The time-lapse results of the flooding experiments with MiliQ water imply that added liquid provides a new surface as no growth was observed beyond the wetting front into the rough terrain (Fig. 5D). Bacteria at the wetted front were stuck in place, with a very limited exploration, they formed wrinkles similar to the edge of the mother colony. Bacteria between the mother colony and the new wetted front were able to grow locally, they filled the area completely, and there was no extensive vertical growth or wrinkle formation. In an analogy with road aquaplaning where a layer of water builds between the surface and the tire, a water layer between the bacterium and surface results in a loss of traction that allows bacteria to spread across the otherwise rough surface essentially removing the underneath roughness.

This is very different from the case of wetted rough agar surfaces where available water acts as a lubricant for bacterial mobility but does not remove surface irregularities. To check if available water limits the expansion on rough surfaces, we have measured the water activity of different agars. The results of water activity measurement (Fig. S7 in Supporting Information Appendix) show a linear decrease in water activity with increasing agar concentrations. The decrease, however, was very small ($a_w$ decreased from 0.9935 to 0.9917 from 0.75% to 6% agar, respectively). Such a minor difference in available water cannot explain a major change in the exploratory behavior observed.

## DISCUSSION

How bacteria move across semi-solid surfaces and conquer new territory is arguably the least understood aspect of bacterial mobility. In part, this is because the process does not depend only on biological drivers (i.e., mode of bacterial motility such as swimming, gliding, or twitching) but is largely determined by physical and mechanical properties of the surface. In addition, it is very challenging to determine the friction coefficient between highly deformable bacterial colonies and semi-solid surface materials. Here, we show how bacterial expansion depends on biofilm viscoelastic properties, and how this is coupled to surface roughness.

On a very smooth agar surface with asperities on the nanometer scale, bacteria ease the spreading by excreting extracellular polymers and surfactin (7, 27–29), which lubricate and decrease the surface tension between the agar and bacteria. The secreted extracellular material is also responsible for the cohesive energy between bacteria, which allows bacteria to spread over the surface as an ensemble (44). It has been shown that hyperflagellated *B. subtilis* cells at the forefront of the dendrite can move as an ensemble (39). Such a strategy, however, has its drawbacks. As we have demonstrated, the overproduction of the adhesive extracellular material increases the cohesive energy in the colony to the point which ultimately results in the bacterial inability to move even on smooth surfaces. It was previously shown that the production of EpsE in *B. subtilis*, which acts as a clutch on the flagella rotor, can inhibit motility (45).

On rough agar surfaces, local stick-slip events are altered by the energy dissipation due to the interaction of bacterial cells with larger surface asperities. As surface friction increases proportionally to the increase in surface roughness, the ability to explore new territory is blocked altogether (46). Unable to move horizontally, the growing *B. subtilis* resolve the increased internal stress in the mother colony through mechanical instabilities which result in wrinkle formation (12). Vertical growth in the mother colony eventually becomes nutrient-limited, bacteria stop growing and enter the stationary phase, which is the end of the growth cycle of bacterial colonies on rough surfaces and gives the bacterial colony its finite size. The most dramatic observation of bacterial frustration to horizontally explore new territories on rough surfaces is observed at the

edge of the mother colony where the bacterial colony rises at a very sharp angle from the agar surface. To maintain such a steep slope, cohesive energy between bacterial cells must be strong enough to resist the force of gravity. Increased viscoelasticity and cohesive energy of biofilms grown on rough surfaces explain why such a steep slope can be maintained. This shows the pragmatic nature of bacterial growth. Because bacteria cannot move laterally due to mechanical constraints they grow vertically to produce colonies of finite sizes. Most of our knowledge about the growth of bacteria on semi-solid surfaces and bacterial colonies comes from experiments on 1.5% agar. This, however, has a degree of bias when we interpret bacterial growth and behavior. Large defined bacterial colonies such as those used in the lab for counting bacteria, or for isolating pure bacterial cultures, maintaining bacterial cultures, genetic recombination, or other standard bacterial lab procedures are rarely observed in natural environments.

The potential for unbounded long-range exploration is an inherent property of bacterial growth (7). It is therefore natural for bacteria to avoid crowding and to try to conquer as much territory as possible as quickly as conditions permit. There are, however, obvious mechanical and physio-chemical constraints that limit the unbound expansion (47). Finding smooth soft surfaces in the environment is not common. Given that in the natural environments nutrient supplies are limited and scattered, hot spots and hot moments of bacterial activity are rare and confined to small spatiotemporal scales which is a severe limitation for long-range expansion. In addition, the gliding surface should be wet during the entire exploration phase which is difficult to maintain in an open environment for a long period of time. Nevertheless, when conditions are right, as during flooding, dispersion and fast bacterial growth rate will inevitably be linked with fast colonization of new territory, a hallmark of bacterial evolutionary success.

We have shown that bacteria have different gene expressions on different agar surfaces. This is an interesting fact that deserves further study in the future. By altering the expression of the extracellular matrix components on different agars, bacteria change the viscoelasticity and cohesive energy of the biofilm and consequently, their ability to explore new territory. In extreme cases such as on rough surfaces, bacteria increase intercellular cohesion to the point that stops horizontal exploration of the new territory. It is currently unknown how bacteria detect surface roughness and/or stiffness. It is obvious, however, that whatever the bacterial proprioception mechanism, the downstream signaling events can lead to a change in gene expression that tunes the extent of the exploration growth. It is also not known if this strategy has a survival advantage for the colony. The results explicitly suggest that when an opportunity arises, the potential for exploration in a stalled bacterial colony is quickly restored and bacteria maximize their growth territory by laterally expanding the colony. To do so, they need to modify two fundamental physical parameters as follows: viscosity of the bacterial colony and local interfacial forces.

## MATERIALS AND METHODS

### Bacterial strains

*Bacillus subtilis* PS-216 wt and its derivative strains were used in this study (Table S1 in Supporting Information Appendix). Mutant strains Δ*epsA-O* (Tc), Δ*sinR* (Phleo), and Δ*srfA* (*srfA::Tn917*) (Mls) were obtained by transforming the parent strain PS-216 with chromosomal DNA obtained from strain *B. subtilis* DL4300, IS720, and ZK722, respectively (Table S1 in Supporting Information Appendix).

*B. subtilis* strain PS-216 is a natural isolate of *B. subtilis*, obtained from the sandy soil samples on the bank of the River Sava (48). The chromosomal DNA (or plasmid) was introduced to the *B. subtilis* using standard transformation protocol, and transformants were plated on LB (Luria-Bertani) agar plates with the appropriate antibiotics: Tc (10 µg/mL), Phleo (1 µg/mL), and Mls [erythromycin (0.5 µg/mL) and lincomycin (12.5 µg/mL)].

The *mKate2* and P*epsA* sequences were amplified using P3F/P3R (49) and P5F/P5R (50) primer pairs, respectively. The amplified *mKate2* sequence was then digested with *HindIII*

and *EcoRI* enzymes, while the P$_{epsA}$ sequence was digested with the *BamHI* and *EcoRI* restriction enzyme pair and ligated in previously digested P$_{sacA}$::P$_{43}$-*yfp* (51). The vector was digested with both *HindIII*, *EcoRI* and *BamHI*, *EcoRI* restriction enzymes. This allows integration of P$_{epsA}$ into the P$_{43}$ locus and replacement of the *yfp* gene with *mKate2* and generation of pMS11 plasmid. Plasmid was then transformed into BM1454 and selected for Cm and Sp.

Bacterial strains were stored at −80°C. The strains were transferred to LB solid agar plates [tryptone 10 g/L, yeast extract 5 g/L, NaCl 5 g/L, and agar 1.5 g/L (wt/vol)] prior to experiments and incubated for 24 h at 37°C. Overnight cultures were grown in liquid LB medium (tryptone 10 g/L, yeast extract 5 g/L, and NaCl 5 g/L) containing tetracycline (10 µg/mL), phleomycin (1 µg/mL), erythromycin (0.5 µg/mL), and lincomycin (12.5 µg/mL) at 37°C with shaking (200 rpm) for 16 h.

## Growth media and growth conditions

Bacteria were grown on MSgg agar plates [100 mM MOPS (3-(N-morpholino)propane sulfonic acid), 5 mM K$_3$PO$_4$, 50 mg/L tryptophan, 50 mg/L phenylalanine, 2 mM MgCl$_2$ · 6H$_2$O, 0.5% (wt/vol) sodium glutamate, 0.5% (wt/vol) glycerol, 700 µM CaCl$_2$ · 2H$_2$O, 50 µM FeCl$_3$ · 6H$_2$O, 50 µM MnCl$_2$, 1 µM ZnCl$_2$, 2 µM thiamine hydrochloride; pH = 7] with various agar [0.75%–6% (wt/vol)] and nutrient concentration [25%–100% (vol/vol)]. To obtain agar plates with different nutrient concentrations, the standard MSgg medium ($C_N$ = 100%) was diluted with MiliQ water. Twenty-five milliliters of medium was poured into Petri dishes with a diameter of 90 mm and air-dried. To monitor the morphology of bacterial strains, 1 µL of overnight culture was inoculated on agar plates with different agar and/or nutrient concentrations and incubated in climatic chamber ICH260L (Memmert, Germany) at 37°C, 80% RH (relative humidity) for 20 h. After the incubation, the morphology was observed by Leica stereomicroscope MZ FLIII and with optical interferometry.

## Viscoelasticity

To measure the viscoelastic properties of MSgg agar plates with various agar concentrations and biofilms that were grown on agar plates with different agar concentrations, we used a modular rotation rheometer Anton Paar Physica MCR 302 with a plate-plate measuring system. Samples were measured at 25.00°C with the plate-plate system PP25 (diameter 24.979 mm), and the gap between the measuring system and the sample was set by determining the normal force ($F_N$), which was 0.25 N ± 0.3 N. To determine the viscoelastic properties of biofilms, 220 µL of the overnight bacterial cultures was spread on the agar plates (90 mm) with different agar concentrations. The samples were incubated in a climatic chamber ICH260L (Memmert, Germany) at 37°C, 80% RH humidity for 20 h. To determine the viscoelastic response of agars, a circle with a diameter of 25 mm was cut from the agar plates and measured. The confluent biofilms grown on different agar concentrations and ≈80 µL were scraped from the agar plates with an object slide and applied to the measuring system. The viscoelasticity of different agars and biofilms was measured with the Amplitude Sweep method, at a constant frequency (10 rad/s). The 25 logarithmically spaced measuring points were captured, and during this time the strain increased from 0.01% to 100%. The yield and flow point were determined from the viscoelastic curves.

## Extent of long-range expansion

To determine the extent of long-range expansion, we monitored the growth of *B. subtilis* PS-216 wt strain on a semi-solid Msgg medium with an agar concentration of 0.75% (wt/vol). The growth was monitored in a round Petri dish with a diameter of 90 mm, and in plastic tubes of length 300 mm and 1,200 mm. One microliter of the overnight bacterial culture was inoculated at the center of the Petri dish or at one end of the tube. Into tubes, the MSgg medium was added using a Pasteur pipette to ≈1/4 of the volume

of the tube. Samples were incubated in a climatic chamber ICH260L at 37°C, 80% RH humidity for 20 h, with the exception of the longest tube, which did not fit into the dimensions of the chamber. This tube was incubated at 37°C surrounded with wet paper towels to maintain high humidity. After 20 h of incubation, samples were photographed with a Canon EOS 600D digital camera.

## Removing expansion barriers

To determine if the addition of surfactin can remove the expansion barrier, 1 µL of an overnight culture of the PS-216 ΔsrfA strain, which has impaired surfactin production, was inoculated in the center of the Petri dishes with 90 mm diameter. Samples were incubated in the climatic chamber ICH260L (Memmert, Germany) at 37°C, 80% RH humidity for 20 h. After 20 h of incubation, 15 µL of either surfactin sodium salt solution [1 mg/mL (wt/vol)] or MiliQ water was added at a single point or around the mother colonies and again incubated in a climatic chamber ICH260L at 37°C, 80% RH humidity for another 20 h. The morphology and the diameter of the enlarged colonies were determined by Leica stereomicroscope MZ FLIII.

## Expression of *epsA-O* operon and *srfA* gene

One microliter of an overnight culture of *B. subtilis* PS-216 $P_{srfAA}$ $P_{epsA}$ was inoculated on agar plates with different agar concentrations and incubated in climatic chamber ICH260L (Memmert, Germany) at 37°C, 80% RH humidity for 20 h. After incubation, biofilms were scraped from the plates with an inoculation loop and transferred to 1 mL of saline solution. The samples were sonicated with a sonotrode for 5 s with 12-micron amplitude using an ultrasonic disintegrator with a 3 mm exponential tip (MSE Scientific Instruments, UK). To measure the expression of *epsA-O* and *srfA* gene, samples were allocated to four technical replicates in the wells of a 96-well black transparent-bottom microtiter plate (Cellstar, Greiner Bio-One, Austria) and placed into a Cytation 3 imaging reader (BioTek, USA). To monitor the $P_{srfAA}$-*yfp* expression, YFP fluorescence intensity with excitation at 500 nm and emission at 530 nm was measured. To monitor the $P_{epsA}$-*mKate2* expression, we measured *mKate2* (red fluorescent protein) fluorescence intensity with excitation at 570 nm and emission at 620 nm.

## Determination of biofilm's biomass

One microliter of an overnight culture of *B. subtilis* PS-216 wt strain was inoculated on agar plates with different agar concentrations and incubated in climatic chamber ICH260L (Memmert, Germany) at 37°C, 80% RH humidity for 20 h. After incubation, the resulting biofilms were scraped from the plates with an inoculation loop and transferred to pre-weighed microcentrifuges. The samples were then dried in an oven (Binder, USA) for 24 h at 55°C to a constant weight. After drying, the samples were weighed on an analytical scale.

## Water activity

For water activity measurements, a circle-shaped agar sample was cut from the agar plates, placed on the carrier, and measured with a water activity meter AquaLab 3TE (Meter, Germany).

## Topography

The surfaces of the samples were analyzed with a 3D optical microscope (ContourGTK0, Bruker Corporation, Billerica, MA, USA) using an interferometric white light objective with 5× magnification. Profilometry is based on scanning white-light interferometry, in which the distance between the sample and the interferometric objective is automatically varied while the corresponding micrographs showing the vertical displacement of interference fringes are recorded. The surface data generated by the profilometer were

processed and analyzed using Bruker's proprietary Contour software. Two-dimensional surface profiles were taken at various locations to determine the topographic characteristics of features observed, namely the height and the length of various biofilms, as well as the topography of different agar samples.

## Roughness

Roughness parameters were measured using the same 3D optical microscope as the general topography. Four common surface roughness areal (3D) parameters were evaluated (ISO 21920-2:2021). Two of them are height parameters: $S_a$, which is the arithmetical mean of the absolute values of the profile deviations from the mean line of the roughness profile, evaluated over the measured surface, and $S_q$ representing the RMS (root mean square) value of the profile, namely the standard deviation of the surface heights, evaluated over the surface. The other two parameters are the moments of amplitude distribution parameters. $S_{sk}$ is skewness, a measure of the asymmetry of the profile about the mean line, but evaluated over the surface, and $S_{ku}$, which is kurtosis, representing a measure of the peakedness of the profile about the mean line, also evaluated over the measured surface.

## Continuum modeling

To model the bacterial expansion of *B. subtilis*, we developed a continuum model that considers the growth and mechanics of bacterial colonies. We used a phase field $\phi$ to track the regions that are occupied by bacterial colonies ($\phi = 1$) versus regions that are devoid of bacteria ($\phi = 0$) (52, 53). The time evolution of $\phi$ is given as follows:

$$\partial_t\phi + u \cdot \nabla\phi = \Gamma\left[\varepsilon\nabla^2\phi + \varepsilon^{-1}G'(\phi) + \varepsilon\kappa_\phi|\nabla\phi|\right], \tag{1}$$

where the term $u \cdot \nabla\phi$ describes the advection of $\phi$ by the expansion velocity $u$ of the colony, $G(\phi) = 18\phi^2(1 - \phi^2)$ is a double-well potential that ensures $\phi$ to be either 1 or 0 in the bulk regions, $\Gamma$ is the relaxation rate, $\varepsilon$ is the width of the interface between $\phi = 1$ and $\phi = 0$, and $\kappa_\phi = -\nabla \cdot (\nabla\phi/|\nabla\phi|)$ denotes the local curvature. Using the phase-field formulation, we model the growth of the colony as follows:

$$\partial_t(\phi\rho) + \nabla \cdot (\phi\rho u) = g_\rho\phi\rho(1 - \rho/\rho_{\max}), \tag{2}$$

where $\rho$ is the local bacterial density, $g_\rho$ is the growth rate, and $\rho_{\max}$ denotes the carrying capacity of the colony. To obtain the expansion velocity $u$, we consider the balance of four forces: (1) a cell-substrate friction $f_{\text{fric}}$, (2) a growth-induced active pressure $p_\rho$, (3) a viscous stress inside a colony $\sigma_{\text{vis}}$, and (4) a Marangoni force $f_M$ due to gradient of surface tension, which yields

$$f_{\text{fric}} - \nabla p_\rho + \nabla \cdot \sigma_{\text{vis}} + f_M = 0. \tag{3}$$

The cell-substrate friction is modeled as a drag force $f_{\text{fric}} = -\xi u$ with a friction coefficient $\xi$. The growth-induced pressure $p_\rho = \eta\phi\rho$ is modeled to be proportional to the density $\rho$ with a proportionality constant $\eta$. The viscous stress inside the colony is given by $\sigma_{\text{vis}} = \upsilon\phi(\nabla u + \nabla u^T)$ where $\upsilon$ denotes the viscosity of the colony. To model the surface tension gradient, we consider the surfactin concentration $c$ that follows the Laplacian equation $\nabla^2 c = 0$ with a high concentration $c = 1$ inside the colony $\phi \gt 0.9$ and a low concentration $c = 0$ far away from the colony $\phi \lt 0.01$. To connect the surfactin concentration $c$ to the Marangoni force $f_M$, we used the following relation $f_M = f_0\tanh|\nabla c|\left(1 + \tanh\dfrac{|\nabla c| - \beta_0}{\delta}\right)$ (54), where $f_0$ is the half maximum of $f_M$, and $\beta_0$ and $\delta$ are the parameters of the sigmoidal increase of $f_M$ with

the concentration gradient $|\nabla c|$. We solved equations (1–3) numerically on a discretized square grid of 256 × 256 points. Equations (1) and (2) were solved using a forward Euler scheme with a fixed time increment $\Delta t = 5 \times 10^{-4}$. The force-balance equation was solved using a semi-implicit Fourier-spectral method as described previously (55). To explore the effect of nutrient and agar concentration on the expansion, we varied the values of $g_\rho$, which presumably increases with nutrient concentration, and the values of $\xi$ and $\upsilon$, which presumably increases with agar concentration. Other model parameters were fixed by $\Gamma = 1$, $\varepsilon = 0.5$, $\rho_{max} = 10$, $\eta = 2$, $f_0 = 10$, $\beta_0 = 2.4$, and $\delta = 0.6$.

## ACKNOWLEDGMENTS

The work was supported by the Slovenian Research Agency (ARRS) national program grant P4-0116, P2-0231, L7-3186 research grant, as well as ARRS 53622 young investigator grant.

The authors thank Hana Prtenjak and Dr. Muhammad Shahid Arshad for their help in performing the topography analysis.

All the listed authors have substantially contributed to the concept and design of the work, revised it critically, approved the final version, and are accountable for the integrity of the work and appropriate investigation.

## AUTHOR AFFILIATIONS

[1]Biotechnical Faculty, Department of Microbiology, University of Ljubljana, Ljubljana, Slovenia

[2]Lewis-Sigler Institute for Integrative Genomics, Carl C. Icahn Laboratory, Princeton University, Princeton, New Jersey, USA

[3]Department of Mechanical and Aerospace Engineering, Princeton University, Princeton, New Jersey, USA

[4]Princeton Materials Institute, Princeton University, Princeton, New Jersey, USA

[5]Faculty of Mechanical Engineering, University of Ljubljana, Ljubljana, Slovenia

## AUTHOR ORCIDs

Mojca Krajnc  http://orcid.org/0009-0006-4292-4780
David Stopar  http://orcid.org/0000-0002-4222-1333

## FUNDING

| Funder | Grant(s) | Author(s) |
| --- | --- | --- |
| Javna Agencija za Raziskovalno Dejavnost RS (ARRS) | P4-0116 | David Stopar |
| Javna Agencija za Raziskovalno Dejavnost RS (ARRS) | P2-0231 | Mitjan Kalin |
| Javna Agencija za Raziskovalno Dejavnost RS (ARRS) | L7-3186 | David Stopar |
| Javna Agencija za Raziskovalno Dejavnost RS (ARRS) | 53622 | Mojca Krajnc |

## AUTHOR CONTRIBUTIONS

Mojca Krajnc, Conceptualization, Formal analysis, Investigation, Methodology, Visualization, Writing – original draft, Writing – review and editing | Chenyi Fei, Conceptualization, Formal analysis, Software, Visualization, Writing – review and editing | Andrej Košmrlj, Conceptualization, Supervision, Writing – review and editing | Mitjan Kalin, Conceptualization, Data curation, Formal analysis, Methodology, Supervision, Visualization, Writing – review and editing | David Stopar, Conceptualization, Funding acquisition, Supervision, Validation, Writing – original draft, Writing – review and editing

## ADDITIONAL FILES

The following material is available online.

## Supplemental Material

**Supplemental material (Spectrum02740-23-s0001.docx).** Fig. S1 to S7 and Table S1.

## Open Peer Review

**PEER REVIEW HISTORY (review-history.pdf).** An accounting of the reviewer comments and feedback.

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
