## [Reviewer comments · Microbiology Spectrum]

Microbiology Spectrum

Mechanical Constraints to Unbound Expansion of *B. subtilis* on Semi-Solid Surfaces

Mojca Krajnc, Chenyi Fei, Andrej Košmrlj, Mitjan Kalin, and David Stopar

Corresponding Author(s): David Stopar, Univerza v Ljubljani Biotehniška fakulteta

Review Timeline:

Submission Date:	July 6, 2023
Editorial Decision:	September 6, 2023
Revision Received:	September 18, 2023
Editorial Decision:	October 10, 2023
Revision Received:	October 11, 2023
Accepted:	October 13, 2023

Editor: Ilana Kolodkin-Gal

Reviewer(s): Disclosure of reviewer identity is with reference to reviewer comments included in decision letter(s). The following individuals involved in review of your submission have agreed to reveal their identity: Tatyana L. Povolotsky (Reviewer #1); Roberto Grau (Reviewer #2)

Transaction Report:

DOI: <https://doi.org/10.1128/spectrum.02740-23>

September 6, 2023

Prof. David Stopar
Univerza v Ljubljani Biotehniška fakulteta
Ljubljana 1000
Slovenia

Re: Spectrum02740-23 (Mechanical Constraints to Unbound Exploratory Growth of *B. subtilis* on Semi-Solid Surfaces)

Dear Prof. David Stopar:

Thank you for submitting your manuscript to Microbiology Spectrum. When submitting the revised version of your paper, please provide (1) point-by-point responses to the issues raised by the reviewers as file type "Response to Reviewers," not in your cover letter, and (2) a PDF file that indicates the changes from the original submission (by highlighting or underlining the changes) as file type "Marked Up Manuscript - For Review Only." Please use this link to submit your revised manuscript - we strongly recommend that you submit your paper within the next 60 days or contact me. Detailed instructions on submitting your revised paper are below.

Link Not Available

Below you will find instructions from the Microbiology Spectrum editorial office and comments generated during the review. Please address carefully the concerns of both reviewers, with complete answers for the justified questions of reviewer 2

Sincerely,

Ilana Kolodkin-Gal

Journals Department
Reviewer comments:

Reviewer #1 (Comments for the Author):

In the manuscript the authors set out to elucidate bacteria's ability to explore and colonize its environment. They investigated the effect of surface mechanical properties (as defined by the differing viscoelastic properties of the agar surface that translated into surface roughness and topology) on *B. subtilis*'s ability to spread over the agar surface. They also investigated the contribution of biofilm and surfactant production on the ability of *B. subtilis* to explore and colonize its immediate environment. Finally, they explored whether proposed genetic deletions could be overcome by external addition of surfactants or water through the phenomenon of aquaplaning. Overall, by examining the viscoelastic properties of the bacterial biofilm and the corresponding relationship to surface stiffness, the manuscript sheds new insights into how the long-range exploratory bacterial growth occurs and that it is intimately dependent on the physical properties of its immediate environment. The manuscript is well written and logically structured. The experiments are technically sound and the results support the presented conclusions of the work. The research would be of interest to the scientific community, is novel and is in Microbiology Spectrum's scope.

Reviewer #2 (Comments for the Author):

The authors present a characterization of the physical and abiotic parameters (roughness, roughness, extracellular matrix components and viscoelasticity of the biofilm) that affect the surface-associated mobility of the model bacterium *Bacillus subtilis*. This behavior is monitored as the concentrations of agar in the culture medium and the supply of nutrients change. The work is interesting from the basic point of view to begin to understand the way in which factors of the extracellular environment affect the ability of a bacterium to move. The work is well written, it is understandable both for a non-expert in Microbiology and for a non-expert in Physics. However, there are some "minor" points to clarify.

1- Is the *B. subtilis* strain used by the authors (PS-216 strain) derived from the NCIB3610 strain? The reference (reference #1) that the authors give for this strain corresponds to a review on different forms of bacterial mobility and I could not find any mention of the *B. subtilis* PS-216 strain. Could the authors indicate the origin of this strain or give an appropriate reference?

2- If the PS-216 strain derives from the NCIB3610 strain or another non-domesticated strain, it would be a prototrophic strain, so why to work with the MSgg medium with the Trp and Phe amino acid aggregates that are necessary for the growth of the domesticated strain JH642 but not for strain NCIB3610 or strains derived from it. In line 98 it should not be indicated that it is a minimal medium (if the mentioned amino acids are not essential for the growth of the strain used in this work).

3- The use of the term "exploratory" is not clear to me since it could give the idea that it is a "back and forth" movement of the bacterium from the colony ("out") and back ("back") to it after exploring the territory. The closest thing that comes to mind now would be the adventurous movement (A motility) of *Mixococcus xanthus* as opposed to its social movement (S Motility). Perhaps the use of the word "exploratory" could be replaced with something else like "social" or "multicellular" movement.

4- In fig S4, it is shown that the production of surfactin and EPS is maximum when the colony (solid biofilm) develops on MSgg plates with agar at high concentrations (6%). Since the motility mechanism described by the authors (although they do not state it explicitly) would be dependent on the flagellum (swarming motility) at this high concentration of agar (4 - 6%), the overproduction of EPS and surfactin could indicate one or two alternatives: biofilm formation is favored (which requires surfactin and EPS) and/or the social mechanism of sliding displacement on surfaces would be activated at later times, to which the authors do not refer. Another alternative, not exclusive of the above, could be that the overproduction of EPS and surfactin would allow several layers of cells to form, one on top of the other, as opposed to the monolayer of mobile cells on a surface with a low concentration of agar (0.75%) or other types of obstacles. A mention should be made to these alternatives between biofilm formation and social sliding.

A minor point: throughout the entire manuscript "the space" is missing between the word and the "(" where the references are indicated (for example in lines 54, 55, 59, 63, 64, ..., 316, 319, ...).

Staff Comments:

Preparing Revision Guidelines

Please return the manuscript within 60 days; if you cannot complete the modification within this time period, please contact me. If you do not wish to modify the manuscript and prefer to submit it to another journal, please notify me of your decision immediately so that the manuscript may be formally withdrawn from consideration by Microbiology Spectrum.

Point by point reply to reviewers' comments

Reviewer #1 (Comments for the Author):

*In the manuscript the authors set out to elucidate bacteria's ability to explore and colonize its environment. They investigated the effect of surface mechanical properties (as defined by the differing viscoelastic properties of the agar surface that translated into surface roughness and topology) on *B. subtilis*'s ability to spread over the agar surface. They also investigated the contribution of biofilm and surfactant production on the ability of *B. subtilis* to explore and colonize its immediate environment. Finally, they explored whether proposed genetic deletions could be overcome by external addition of surfactants or water through the phenomenon of aquaplaning. Overall, by examining the viscoelastic properties of the bacterial biofilm and the corresponding relationship to surface stiffness, the manuscript sheds new insights into how the long-range exploratory bacterial growth occurs and that it is intimately dependent on the physical properties of its immediate environment. The manuscript is well written and logically structured. The experiments are technically sound and the results support the presented conclusions of the work. The research would be of interest to the scientific community, is novel and is in Microbiology Spectrum's scope.*

We would like to thank the reviewer for the encouraging words.

Reviewer #2 (Comments for the Author):

*The authors present a characterization of the physical and abiotic parameters (roughness, roughness, extracellular matrix components and viscoelasticity of the biofilm) that affect the surface-associated mobility of the model bacterium *Bacillus subtilis*. This behavior is monitored as the concentrations of agar in the culture medium and the supply of nutrients change. The work is interesting from the basic point of view to begin to understand the way in which factors of the extracellular environment affect the ability of a bacterium to move. The work is well written, it is understandable both for a non-expert in Microbiology and for a non-expert in Physics. However, there are some "minor" points to clarify.*

We appreciate that the reviewer finds our work “interesting” and “well-written”. We would also like to thank the reviewer for providing constructive feedbacks, which helped us improve the manuscript.

*1- Is the *B. subtilis* strain used by the authors (PS-216 strain) derived from the NCIB3610 strain? The reference (reference #1) that the authors give for this strain corresponds to a review on different forms of bacterial mobility and I could not find any mention of the*

B. subtilis PS-216 strain. Could the authors indicate the origin of this strain or give an appropriate reference?

B. subtilis strain PS-216 is not a derivative of NCIB3610 strain. It is a natural isolate of *B. subtilis*, obtained from the sandy soil samples on the bank of the River Sava, Slovenia (grid reference 46° 06' N, 14° 28' E) in January 2006. The bacterial strain was first described in the work of Stefanic P, Mandic-Mulec I. 2009 entitled "Social interactions and distribution of *Bacillus subtilis* phenotypes at microscale" in *J Bacteriol* 191:1756–1764 (Supplementary Ref. 1 of the original manuscript) and has been used ever since in our laboratory.

Revisions: We have cited the paper mentioned above in the revised manuscript and clearly stated the origin of the strain in the main text.

Lines 367-368: *B. subtilis* strain PS-216 is a natural isolate of *B. subtilis*, obtained from the sandy soil samples on the bank of the River Sava (49).

2- *If the PS-216 strain derives from the NCIB3610 strain or another non-domesticated strain, it would be a prototrophic strain, so why to work with the MSgg medium with the Trp and Phe amino acid aggregates that are necessary for the growth of the domesticated strain JH642 but not for strain NCIB3610 or strains derived from it. In line 98 it should not be indicated that it is a minimal medium (if the mentioned amino acids are not essential for the growth of the strain used in this work).*

B. subtilis strain PS-216 is a non-domesticated strain and can grow on a variety of substrates (i.e. LB, MSgg, TY, B medium). We have selected MSgg growth medium as it is often used in studies of *B. subtilis* biofilms (e.g., doi: <https://doi.org/10.1128/spectrum.00908-23>; <https://doi.org/10.1111/j.1365-2958.2007.06040.x>; <https://doi.org/10.3389/fmicb.2018.00105>; <https://doi.org/10.1111/mmi.14127>). We agree with the reviewer that MSgg is not a minimal medium for strain PS-216.

Revision: In line 109 of the revised manuscript, we have removed "minimal" as suggested by the reviewer.

3- *The use of the term "exploratory" is not clear to me since it could give the idea that it is a "back and forth" movement of the bacterium from the colony ("out") and back ("back") to it after exploring the territory. The closest thing that comes to mind now would be the adventurous movement (A motility) of *Mixococcus xanthus* as opposed to its social movement (S Motility). Perhaps the use of the word "exploratory" could be replaced with something else like "social" or "multicellular" movement.*

We thank the reviewer for this suggestion. We agree that exploratory growth might be ambiguous in the sense explained by the reviewer.

Revisions: To make this clear we have replaced exploratory growth with expansion throughout the text. The expansion describing spreading of bacteria on solid surfaces has been used in the literature before (e.g., doi: [10.1039/d0sm01348j](https://doi.org/10.1039/d0sm01348j)).

4- In fig S4, it is shown that the production of surfactin and EPS is maximum when the colony (solid biofilm) develops on MSgg plates with agar at high concentrations (6%). Since the motility mechanism described by the authors (although they do not state it explicitly) would be dependent on the flagellum (swarming motility) at this high concentration of agar (4 - 6%), the overproduction of EPS and surfactin could indicate one or two alternatives: biofilm formation is favored (which requires surfactin and EPS) and/or the social mechanism of sliding displacement on surfaces would be activated at later times, to which the authors do not refer. Another alternative, not exclusive of the above, could be that the overproduction of EPS and surfactin would allow several layers of cells to form, one on top of the other, as opposed to the monolayer of mobile cells on a surface with a low concentration of agar (0.75%) or other types of obstacles. A mention should be made to these alternatives between biofilm formation and social sliding.

We thank the reviewer for bringing up the different possibilities.

Our results show that bacteria try to overcome mechanical constraints on higher agar concentrations by increased production of surfactin. Whereas surfactin should facilitate colony expansion, the concomitant increase in EpsA-O polysaccharide promotes biofilm formation and inhibits colony expansion. The results on higher agar surfaces suggest that biofilm formation prevails, as the colonies are unable to expand laterally under such conditions (Fig. 1) and can only grow vertically to form structures that are several cell layers thick (Fig. 4c), as opposed to the monolayer structures formed by swarming cells on a surface with a low concentration of agar (Fig. 4a). This may also provide a mechanical explanation why researchers have been studying swarming motility at relatively low agar concentrations (between 0.3 % to 1.0 %; see e.g., doi: [10.1038/nrmicro2405](https://doi.org/10.1038/nrmicro2405)).

We did not think the restricted colony expansion at high agar concentrations is due to delayed activation of social sliding because we did not observe expansion of colonies on stiff agar even after a prolonged incubation (> 3 days).

Revisions: We have modified the text to bring forward this point more clearly.

Lines 171-175: The increased expression of EpsA-O polysaccharide at higher agar concentrations correlates with biofilm formation several layers thick, as opposed to the monolayer structures of mobile cells on a surface with a low concentration of agar. Even after a prolonged incubation bacterial lateral expansion was halted.

5. A minor point: throughout the entire manuscript "the space" is missing between the word and the "(" where the references are indicated (for example in lines 54. 55. 59, 63, 64,....316, 319,.....

Revisions: We have corrected this in the revised manuscript.

October 10, 2023

Prof. David Stopar
Univerza v Ljubljani Biotehniška fakulteta
Ljubljana 1000
Slovenia

Re: Spectrum02740-23R1 (Mechanical Constraints to Unbound Expansion of *B. subtilis* on Semi-Solid Surfaces)

Dear Prof. David Stopar:

Two final comments need to be addressed in a minor text revision: In Material and Methods (lines 374 to 377), the names of the used restriction enzymes must be in italics.

- Line 437 (Δ sinR), the "triangle" should not be in italic.
- Line 371, use "lincomycin 12.5 ug/ml" instead "lincomycin 12,5 ug/ml"

Link Not Available

Sincerely,

Ilana Kolodkin-Gal

Journals Department
Reviewer comments:

Reviewer #2 (Comments for the Author):

The authors have answered all concerns satisfactorily. The manuscript is now suitable for publication in Microbiology Spectrum. Minor points,

- In Material and Methods (lines 374 to 377), the names of the used restriction enzymes must be in italics.
- Line 437 (Δ sinR), the "triangle" should not be in italic.
- Line 371, use "lincomycin 12.5 ug/ml" instead "lincomycin 12,5 ug/ml"

Staff Comments:

Preparing Revision Guidelines

Please return the manuscript within 60 days; if you cannot complete the modification within this time period, please contact me. If you do not wish to modify the manuscript and prefer to submit it to another journal, please notify me of your decision immediately so that the manuscript may be formally withdrawn from consideration by Microbiology Spectrum.

Point by point reply to reviewers' comments

Reviewer #1 (Comments for the Author):

*In the manuscript the authors set out to elucidate bacteria's ability to explore and colonize its environment. They investigated the effect of surface mechanical properties (as defined by the differing viscoelastic properties of the agar surface that translated into surface roughness and topology) on *B. subtilis*'s ability to spread over the agar surface. They also investigated the contribution of biofilm and surfactant production on the ability of *B. subtilis* to explore and colonize its immediate environment. Finally, they explored whether proposed genetic deletions could be overcome by external addition of surfactants or water through the phenomenon of aquaplaning. Overall, by examining the viscoelastic properties of the bacterial biofilm and the corresponding relationship to surface stiffness, the manuscript sheds new insights into how the long-range exploratory bacterial growth occurs and that it is intimately dependent on the physical properties of its immediate environment. The manuscript is well written and logically structured. The experiments are technically sound and the results support the presented conclusions of the work. The research would be of interest to the scientific community, is novel and is in Microbiology Spectrum's scope.*

We would like to thank the reviewer for the encouraging words.

Reviewer #2 (Comments for the Author):

*The authors present a characterization of the physical and abiotic parameters (roughness, roughness, extracellular matrix components and viscoelasticity of the biofilm) that affect the surface-associated mobility of the model bacterium *Bacillus subtilis*. This behavior is monitored as the concentrations of agar in the culture medium and the supply of nutrients change. The work is interesting from the basic point of view to begin to understand the way in which factors of the extracellular environment affect the ability of a bacterium to move. The work is well written, it is understandable both for a non-expert in Microbiology and for a non-expert in Physics. However, there are some "minor" points to clarify.*

We appreciate that the reviewer finds our work “interesting” and “well-written”. We would also like to thank the reviewer for providing constructive feedbacks, which helped us improve the manuscript.

*1- Is the *B. subtilis* strain used by the authors (PS-216 strain) derived from the NCIB3610 strain? The reference (reference #1) that the authors give for this strain corresponds to a review on different forms of bacterial mobility and I could not find any mention of the*

B. subtilis PS-216 strain. Could the authors indicate the origin of this strain or give an appropriate reference?

B. subtilis strain PS-216 is not a derivative of NCIB3610 strain. It is a natural isolate of *B. subtilis*, obtained from the sandy soil samples on the bank of the River Sava, Slovenia (grid reference 46° 06' N, 14° 28' E) in January 2006. The bacterial strain was first described in the work of Stefanic P, Mandic-Mulec I. 2009 entitled "Social interactions and distribution of *Bacillus subtilis* phenotypes at microscale" in *J Bacteriol* 191:1756–1764 (Supplementary Ref. 1 of the original manuscript) and has been used ever since in our laboratory.

Revisions: We have cited the paper mentioned above in the revised manuscript and clearly stated the origin of the strain in the main text.

Lines 367-368: *B. subtilis* strain PS-216 is a natural isolate of *B. subtilis*, obtained from the sandy soil samples on the bank of the River Sava (49).

2- *If the PS-216 strain derives from the NCIB3610 strain or another non-domesticated strain, it would be a prototrophic strain, so why to work with the MSgg medium with the Trp and Phe amino acid aggregates that are necessary for the growth of the domesticated strain JH642 but not for strain NCIB3610 or strains derived from it. In line 98 it should not be indicated that it is a minimal medium (if the mentioned amino acids are not essential for the growth of the strain used in this work).*

B. subtilis strain PS-216 is a non-domesticated strain and can grow on a variety of substrates (i.e. LB, MSgg, TY, B medium). We have selected MSgg growth medium as it is often used in studies of *B. subtilis* biofilms (e.g., doi: <https://doi.org/10.1128/spectrum.00908-23>; <https://doi.org/10.1111/j.1365-2958.2007.06040.x>; <https://doi.org/10.3389/fmicb.2018.00105>; <https://doi.org/10.1111/mmi.14127>). We agree with the reviewer that MSgg is not a minimal medium for strain PS-216.

Revision: In line 109 of the revised manuscript, we have removed "minimal" as suggested by the reviewer.

3- *The use of the term "exploratory" is not clear to me since it could give the idea that it is a "back and forth" movement of the bacterium from the colony ("out") and back ("back") to it after exploring the territory. The closest thing that comes to mind now would be the adventurous movement (A motility) of *Mixococcus xanthus* as opposed to its social movement (S Motility). Perhaps the use of the word "exploratory" could be replaced with something else like "social" or "multicellular" movement.*

We thank the reviewer for this suggestion. We agree that exploratory growth might be ambiguous in the sense explained by the reviewer.

Revisions: To make this clear we have replaced exploratory growth with expansion throughout the text. The expansion describing spreading of bacteria on solid surfaces has been used in the literature before (e.g., doi: [10.1039/d0sm01348j](https://doi.org/10.1039/d0sm01348j)).

4- In fig S4, it is shown that the production of surfactin and EPS is maximum when the colony (solid biofilm) develops on MSgg plates with agar at high concentrations (6%). Since the motility mechanism described by the authors (although they do not state it explicitly) would be dependent on the flagellum (swarming motility) at this high concentration of agar (4 - 6%), the overproduction of EPS and surfactin could indicate one or two alternatives: biofilm formation is favored (which requires surfactin and EPS) and/or the social mechanism of sliding displacement on surfaces would be activated at later times, to which the authors do not refer. Another alternative, not exclusive of the above, could be that the overproduction of EPS and surfactin would allow several layers of cells to form, one on top of the other, as opposed to the monolayer of mobile cells on a surface with a low concentration of agar (0.75%) or other types of obstacles. A mention should be made to these alternatives between biofilm formation and social sliding.

We thank the reviewer for bringing up the different possibilities.

Our results show that bacteria try to overcome mechanical constraints on higher agar concentrations by increased production of surfactin. Whereas surfactin should facilitate colony expansion, the concomitant increase in EpsA-O polysaccharide promotes biofilm formation and inhibits colony expansion. The results on higher agar surfaces suggest that biofilm formation prevails, as the colonies are unable to expand laterally under such conditions (Fig. 1) and can only grow vertically to form structures that are several cell layers thick (Fig. 4c), as opposed to the monolayer structures formed by swarming cells on a surface with a low concentration of agar (Fig. 4a). This may also provide a mechanical explanation why researchers have been studying swarming motility at relatively low agar concentrations (between 0.3 % to 1.0 %; see e.g., doi: [10.1038/nrmicro2405](https://doi.org/10.1038/nrmicro2405)).

We did not think the restricted colony expansion at high agar concentrations is due to delayed activation of social sliding because we did not observe expansion of colonies on stiff agar even after a prolonged incubation (> 3 days).

Revisions: We have modified the text to bring forward this point more clearly.

Lines 171-175: The increased expression of EpsA-O polysaccharide at higher agar concentrations correlates with biofilm formation several layers thick, as opposed to the monolayer structures of mobile cells on a surface with a low concentration of agar. Even after a prolonged incubation bacterial lateral expansion was halted.

5. A minor point: throughout the entire manuscript "the space" is missing between the word and the "(" where the references are indicated (for example in lines 54. 55. 59, 63, 64,....316, 319,.....

Revisions: We have corrected this in the revised manuscript.

October 13, 2023

Prof. David Stopar
Univerza v Ljubljani Biotehniška fakulteta
Ljubljana 1000
Slovenia

Re: Spectrum02740-23R2 (Mechanical Constraints to Unbound Expansion of *B. subtilis* on Semi-Solid Surfaces)

Dear Prof. David Stopar:

Your manuscript has been accepted, and I am forwarding it to the ASM Journals Department for publication. You will be notified when your proofs are ready to be viewed.

Sincerely,

Ilana Kolodkin-Gal
Editor, Microbiology Spectrum
